# Availability of Medical Services and Teleconsultation during COVID-19 Pandemic in the Opinion of Patients of Hematology Clinics—A Cross-Sectional Pilot Study (Silesia, Poland)

**DOI:** 10.3390/ijerph20054264

**Published:** 2023-02-27

**Authors:** Kamila Jaroń, Angelika Jastrzębska, Kamil Mąkosza, Mateusz Grajek, Karolina Krupa-Kotara, Joanna Kobza

**Affiliations:** 1Department of Public Health, Department of Public Health Policy, Faculty of Health Sciences in Bytom, Medical University of Silesia in Katowice, Piekarska 18, 41-902 Bytom, Poland; 2Department of Epidemiology, Department of Epidemiology and Biostatistics, Faculty of Health Sciences in Bytom, Medical University of Silesia in Katowice, Piekarska 18, 41-902 Bytom, Poland

**Keywords:** pandemic, COVID-19, SARS-CoV-2, teleconsultation, telemedicine, e-health

## Abstract

Summary: A new virus, SARS-CoV-2, emerged in December 2019, triggering the COVID-19 pandemic in 2020 due to the rapid spread and severity of cases worldwide. In Poland, the first case of COVID-19 was reported on 4 March 2020. The aim of the prevention efforts was primarily to stop the spread of the infection to prevent overburdening the health care system. Many illnesses were treated by telemedicine, primarily using teleconsultation. Telemedicine has reduced personal contact between doctors and patients and reduced the risk of exposure to disease for patients and medical personnel. The survey aimed to gather patients’ opinions on the quality and availability of specialized medical services during the pandemic. Based on the data collected regarding patients’ opinions on services provided via telephone systems, a picture was created of patients’ opinions on teleconsultation, and attention was drawn to emerging problems. The study included a 200-person group of patients, realizing their appointments at a multispecialty outpatient clinic in Bytom, aged over 18 years, with various levels of education. The study was conducted among patients of Specialized Hospital No. 1 in Bytom. A proprietary survey questionnaire was developed for the study, which was conducted on paper and used face-to-face interaction with patients. Results: 17.5% of women and 17.5% of men rated the availability of services during the pandemic as good. In contrast, among those aged 60 and over, 14.5% of respondents rated the availability of services during the pandemic as poor. In contrast, among those in the labor force, as many as 20% of respondents rated the accessibility of services provided during the pandemic as being well. The same answer was marked by those on a pension (15%). Overwhelmingly, women in the age group of 60 and over showed a reluctance toward teleconsultation. Conclusions: Patients’ attitudes toward the use of teleconsultation services during the COVID-19 pandemic varied, primarily due to attitudes toward the new situation, the age of the patient, or the need to adapt to specific solutions not always understood by the public. Telemedicine cannot completely replace inpatient services, especially among the elderly. It is necessary to refine remote visits to convince the public of this type of service. Remote visits should be refined and adapted to the needs of patients in such a way as to remove any barriers and problems arising from this type of service. This system should also be introduced as a target, providing an alternative method of inpatient services even after the pandemic ends.

## 1. Introduction

For a long time, coronaviruses were considered benign pathogens that cause respiratory symptoms of minor severity that resolve within a few days. The arrival of new infectious virus species has given rise to an increase in interest in these viruses. Before the emergence of the new SARS-CoV-2 coronavirus, a highly infectious species of SARS coronavirus had already appeared in the public, in 2002, causing a worldwide outbreak. Ten years after the SARS outbreak, new cases of the respiratory disease caused by the MERS coronavirus emerged, but this virus did not entail an outbreak. In contrast, a new SARS-CoV-2 virus emerged in December 2019, which triggered the COVID-19 pandemic in 2020 due to the rapid spread and severity of cases worldwide [1]. The Wuhan live animal and seafood market is considered the epicenter of COVID-19. In Poland, the first case was reported on 4 March 2020. The aim of the prevention effort was primarily to stem the spread of infection to prevent overburdening the healthcare system [2]. The most common symptoms present at the onset of SARS-CoV-2 coronavirus infection were dry cough, fever, general weakness, and muscle aches. The course of the infection largely depends on the age of the patient, and more severe symptoms are observed more often in the elderly than in children [1]. Most symptomatic patients have a mild form of the disease (80% of patients). In contrast, 14% of symptomatic patients have a severe course of the disease, i.e., accelerated breathing, significant resting dyspnea, involvement of more than 50% of the lung parenchyma, and saturation below 94%. A minority, of 6% of patients have a critical course of the disease with acute respiratory distress syndrome, with multiple organ failure and septic shock [2]. In about 20% of people, the disease is asymptomatic. To a large extent, the course of the disease and its severity depend on the patient’s immune response to infection. Coronavirus, SARS-CoV-2 is primarily transmitted between people by the droplet route, where close person-to-person contact is not necessary. For infection to occur, the virus must be transmitted to the mucous membranes of the throat, nose, or eyes. The minimum infectious dose of the virus has not been determined [3].

The pandemic continues to be a global threat to health care and the availability of health services. It has affected all countries and therefore health systems have had to adapt to the new situation to ensure rapid access to medical care. However, due to reduced access to medical services during this time, the functioning of the healthcare system has been disrupted. To curb the spread of the virus, many diseases were treated through telemedicine, primarily using teleconsultation. Telemedicine has reduced personal contact between doctors and patients and reduced the risk of exposure to disease for patients and medical personnel. However, telemedicine does not fully replace the interaction that occurs in face-to-face interactions [4]. In Poland, the majority of teleconsultations within the framework of so-called telemedicine and medical advice are carried out in contact through a telecommunications device (such as a telephone). According to estimates, this is 95% of all teleconsultations. Other forms, such as video chat, are marginal [2,3,4]. Nonetheless, alternative modes of communication, such as online consultations and teleconsultation, have significant benefits in emergencies. Among other things, they provide patients with real-time information and professional advice from physicians during times of inaccessibility to medical facilities [5].

The purpose of the survey was to gather patients’ opinions on the quality and availability of specialized medical services during the pandemic. Based on the data collected regarding patients’ opinions on services provided via telephone systems, a picture was created of the opinions of clinic patients regarding teleconsultation, and attention was paid to emerging problems. It was assumed that the coronavirus pandemic negatively affected the quality and availability of medical services provided by public health care providers.

## 2. Materials and Methods

### 2.1. Study Organization

The study included a 200-person group of patients, completing their visits to specialized hematology outpatient clinics in Bytom (Silesia, Poland) (Figure 1), aged over 18 years, with various levels of education. To anonymize the study, only data on gender, age, and the fact of treatment were collected. All data were coded with appropriate symbols, preventing the identification of patients by the Act of 29 August 1997, on the Protection of Personal Data (Journal of Laws of 1997, No. 133, item 883).

The primary criteria for inclusion were the patient’s written consent, expressed through participation in the survey, and that the patients be aged 18 or over. Participation in the study was anonymous and completely voluntary. The study adhered to the provisions of the Declaration of Helsinki and received a positive opinion from the Bioethics Committee of the Silesian Medical University in Katowice (ID: PCN/0022/KB/211/20).

### 2.2. Research Tool

A proprietary survey questionnaire was developed for the study, which was conducted on paper and used face-to-face interaction with patients. The survey questionnaire contained 17 closed questions. The first five questions (metric) were about gender, age, place of residence, education, and current occupational status. The remaining 12 questions were aimed at finding out the patients’ opinions on the teleconsultations conducted and assessing their availability and quality. The questionnaire was validated by administering it twice, two weeks apart, to a group of 30 people; in the first version, respondents were given a chance to express their opinion and indicate comments on the content of the questionnaire. The second time, the repetition of responses was tested. The reliability of the questionnaire was assessed using Cronbach’s alpha coefficient and was shown to be 0.83, which in psychological research indicates good reliability.

### 2.3. Study Sample

The study included 200 patients, most of whom were women (58%). The largest number of respondents belonged to the age group of 60 years and older (44%), and the smallest number belonged to the age group of 18–28 years (9%). Of the respondents, 94.5% were city residents and most had a secondary/vocational education (68%). The surveyed patients were mostly employed (50%) or retired (49%) (Table 1).

### 2.4. Statistical Compilation

Statistical analysis was carried out using Statistica software (Statsoft, Poland). Multivariate tables were used in the calculations, individual groups of respondents were compared, and relationships between variables were analyzed. Mann–Whitney U and Kruskal–Wallis tests were used in statistical inference. The *p*-values <0.05 were considered statistically significant. For the results of the statistical inference, the abbreviation T is adopted in the text.

## 3. Results

In response to the question “How do you rate the availability of services provided during the COVID-19 pandemic?”, the majority of respondents rated the availability of services during the COVID-19 pandemic as good (35%), and 25.5% as definitely good. In contrast, 21.5% of respondents marked the answer “difficult to say”, 34 people (17%) rated the availability as bad, and only two people (1%) as definitely bad. To the next question, i.e., “How do you rate the quality of services provided during the COVID-19 pandemic?”, 32% of respondents rated the quality of services provided as good, and 27% of people answered: “hard to say”. Another 20% of respondents rated the quality as good, 15.5% of respondents marked the answer “bad”, while only 5.5% of people answered, “definitely bad”. When asked to evaluate the quality of the services provided through ICT systems, 30.5% of respondents thought that the introduction of teleconsultation and its quality were good. 27.5% had no opinion on the subject, while 21.5% of respondents rated the quality of services provided through ICT systems badly. Of the respondents, 17% gave a decidedly good rating, and 3.5% gave a decidedly bad rating. Furthermore, 56% of respondents indicated that the creation of teleconsultation during the COVID-19 pandemic was a good idea, while 44% indicated that it was not a good idea. In response to the question “What do you like best about the advice provided through telephone or online systems?” (respondents could indicate more than one answer), most respondents indicated the convenience of visiting without leaving home (49.5%), 45.5% marked safety related to the possibility of contracting a virus; however, 45% indicated the answer “I don’t like this type of visit”. Additionally, 30.5% of respondents indicated the lack of waiting in line, while only 17% of people marked the answer that they had better contact with the doctor. In a question about possible problems arising when providing advice via ICT systems (again, it was possible to mark more than one answer), the largest number of people (56%) indicated that they had not noticed any problems in this regard, 40.5% of respondents had problems with connectivity, while 38.5% of people had problems understanding the information provided, 32.5% of respondents indicated poor contact with the doctor, and 26.5% of people indicated a lack of examination. To the question “Do you think it would have been a good idea to conduct visits via ICT systems without the pandemic?”, 54% said yes, while 46% of people indicated a “no” answer. The same number of respondents, as with the previous question, answered the question “Are you willing to use the advice provided by the telephone method?” and 54% indicated “yes”, while 46% indicated “no”. Regarding the question about the attitude of medical personnel to the advice given by the telephone method, 34.5% of respondents answered “difficult to say”, 28.5% of people rated the attitude of medical personnel to the advice given as being well, as did 20.5% of respondents. In contrast, the answer “bad” was marked by 14% of people, and “definitely bad” by 2.5% of respondents. In response to the question “Have you used other medical facilities that also provided telehealth appointments?”, 77.5% of people answered that they had used telehealth elsewhere, while 22.5% of people had not used this type of service elsewhere. The last question included only those who answered yes to the previous question, i.e., “Have you used other medical facilities where teleconsultation visits were also conducted?” and referred to 155 people. This question was about the evaluation of conducted visits to another facility via telehealth systems and 31.6% of people rated the conducted visits to another facility via telehealth systems badly, 29% of people did not comment, 26.5% of respondents rated the visits well, 11% of people marked the answer “definitely badly”, and 1.9% of people marked the answer “definitely well”.

Referring to the question: “How do you rate the availability of services provided during the COVID-19 pandemic?”, a breakdown was made in the responses in terms of the number of women and men (Figure 1). Of men and women, 17.5% rated the availability of provided services during the pandemic well, 15% of women rated this availability strongly well, while only 10.5% of men gave this rating (“strongly well”). A bad rating was given by 12.5% of women and 4.5% of men. The answer “definitely bad” was indicated by 1% of men, and 0% of women. In contrast, 13% of women and 8.5% of men had no opinion. There was no relationship between the variable’s gender and the evaluation of the availability of medical services during the COVID-19 pandemic (*p* > 0.05).

For the same question—How do you rate the availability of services provided during the COVID-19 pandemic?” for respondents by age (Figure 2), in the age group of 60 and over, 14.5% of respondents rated the availability of services provided during the pandemic poorly. The answer good was marked by 4.5% of people, and bad by 1% of respondents. The same number, i.e., 12% of respondents, marked the answer “good” and “hard to say”. In the 50–59 age group, the largest number of respondents answered “good” (7.5% of people). Six percent of respondents marked the answer “definitely good”, and “hard to say” was indicated by 4.5%. No one marked the answers “bad” and “definitely bad”. On the other hand, in the 40–49 age group, the highest number of responses was “good” (7%). “Good” was marked by 4% of respondents, 2.5% of people had no opinion on the subject, and 2% of respondents indicated the answer “bad”. No one marked the answer “definitely bad”. Respondents in the 29–39 age group mostly indicated the answer “definitely good” (5.5%), 5% of people indicated the answer “good”, 0.5% indicated the answer “bad”, and 2.5% had no opinion. Additionally, no one marked the answer “definitely bad”. In contrast, in the 18–28 age group, there are only two ratings, i.e., “definitely good” (5.5%) and “good” (2.5%). A statistically significant relationship was found between the variable age and the evaluation of the availability of services during the pandemic. Those over 60 were more likely to negatively evaluate the availability of medical services provided during the COVID-19 pandemic (T = 11.868; r = 0.632; *p* = 0.001).

About the professional status of the respondents, the answers to the above question—“How would you rate the availability of services provided during the COVID-19 pandemic?” (Figure 3)—were as follows: among working people, as many as 20% of respondents rated the availability of provided services during the pandemic well, 19% of working respondents indicated the answer “definitely well”, 3% “poorly”, while 8% had no opinion. No one marked the answer “definitely bad”. Those on a pension, on the other hand, mostly (15%) marked the answer “good”. Of respondents, 14% marked the answer “bad”, while 13.5% had no opinion on the subject. In contrast, “definitely good” was marked by 5.5% of people, and “definitely bad” by 1%. Those who were pupils or students (1%) marked one answer—”definitely good”. There was a statistically significant relationship between the variable of occupational status and the assessment of the availability of services during the pandemic. Those who were employed/retired were more likely to negatively evaluate the availability of medical services provided during the COVID-19 pandemic (T = 12.003; r = 0.614; *p* = 0.002).

Another question asked “Are you willing to use telephonic advice?”, and respondents were grouped by age and gender (Figure 4). Overwhelmingly, reluctance to teleprompting was shown by women in the age group of 60 years and older (T = 10.099; r = 0.703; *p* = 0.001). The rest of the respondents’ answers were similar, so no differences were noted (*p* > 0.05). The more frequent response was “yes” among both women and men, regardless of age.

## 4. Discussion

The pandemic has changed the way healthcare services are delivered to patients around the world. To provide precautions and physical distancing during the COVID-19 pandemic, telephone consultation was provided as an alternative method to face-to-face visits, primarily in primary care (PCP) [6]. However, telemedicine also has some drawbacks, as it primarily focuses on the symptoms presented by the patient, patients are often not comprehensively examined and visual cues are often lacking. In addition, there are issues regarding the relationship between doctor and patient, or problems regarding the quality of the information provided [6]. Despite the drawbacks, telephone consultations were used during the pandemic because of their ability to deliver remote, essential health care to patients and to halt the spread of the virus [6].

A study by Zammit, et al. found that there was a significant improvement in patient satisfaction and an increased preference for telephone consultations [7]. Telemedicine during the pandemic made a huge impact mainly among older patients or patients with chronic diseases. The advantages of telephone telemedicine, in addition to preventing the transmission of infections, are convenience and saving time. However, the difficulty of checking and explaining the condition to patients, the possibly incomplete assessment of their health status, and the misunderstandings that can arise from a telephone consultation between a doctor and a patient negatively affect this type of medical service [8].

The COVID-19 pandemic has proven that telemedicine is a very helpful and desirable tool in healthcare. It allows for a personalized approach on the part of healthcare professionals toward patients and the establishment of positive interactions between them. This represents a very valuable aspect from the perspective of both parties. The use of telemedicine has made it possible to access medications (so-called e-prescriptions, electronic prescriptions), make diagnoses, implement comprehensive treatment, and, in addition, carry out health education processes, including issues related to the prevention of chronic diseases. Studies related to teleconsultation, which were conducted before the outbreak of the SARS-CoV-2 virus pandemic, did not show a significant decrease in effectiveness compared with traditional visits made in a stationary manner [9,10].

A study that was conducted in the context of the role and importance of telemedicine in the initial wave of the COVID-19 pandemic was the original work carried out by Fatyga et al. [11]. This study was related to elderly patients of a Silesian diabetes clinic. It involved 86 patients, aged ≥60 years, whose leading disease was type 2 diabetes. The study did not include patients with microvascular complications of diabetes, those who had suffered a stroke, were struggling with depression or other mental disorders, or were consuming excessive amounts of alcoholic beverages. The results obtained by the authors show that, for the most part, a significant number of patients—despite complying with all restrictions related to the sanitary-epidemiological regime, i.e., taking preventive behaviors—declared frequent or constant feelings of fear of contracting coronavirus disease. Consequently, alternatives such as the use of telemedicine were far more favorable to them due to the lack of real contact with other people, thereby offsetting the risk of potential illness due to COVID-19. The conclusions of the survey demonstrate the validity of the use of telemedicine, although it is worth considering measures to improve it. In addition, it seems non-negligible to conduct further scientific research, including clinical research, focusing on the issue of telephone and electronic medicine from the point of view of patients, which will allow more accurate interpretations regarding the adequate management of medical personnel in this area, as well as strengthening behavioral health strategies among the elderly population.

Patient satisfaction with the use of telemedicine can also vary depending on the availability of both face-to-face visits and teleconsultation [8]. In a study conducted on the satisfaction and importance of teleconsultation during the coronavirus pandemic among patients with rheumatoid arthritis, 62.3% said the quality of teleconsultation was not as satisfactory when compared with in-person consultations [12]. In contrast, in another study on determining patients’ satisfaction with the quality of teleconsultation. Patients in the surveyed PCPs rated communication with the doctor and comprehensiveness of medical care the highest. The treatment used helped 47.5% of patients improve their health [13]. Additionally, studies have been conducted on the use of telemedicine among asthma patients. However, the disadvantages brought to the fore regarding teleconsultation were the limited ability to perform tests, or the lack of personal contact between doctor and patient [14,15]. From a subsequent study conducted among 14,000 respondents on the satisfaction of patients using teleconsultation with their PCP during the pandemic, more than 40% of respondents were satisfied with the teleconsultation provided and said that the quality of services provided in this way was comparable to the advice given in an inpatient manner. In contrast, 36.3% of people rated the quality of an in-person visit to a PCP higher than a teleconsultation [16]. Thanks to telemedicine, people in high-risk groups, for example those with cardiovascular disease, diabetes, or Parkinson’s disease, were able to effectively monitor their health status during the pandemic, while maintaining constant contact with medical personnel [17].

The study also found that doctors and nurses showed lower satisfaction with teleconsultation than patients. Above all, medical personnel were concerned about emergencies that could occur due to the patient’s limited visualization during a telephone consultation. Telephone consultations tended to convey less information than video consultations; however, despite this, teleconsultation was preferred over video visits by both providers and patients, especially those who were less technologically advanced [8].

The nature of telemedicine may limit a provider’s ability to obtain a comprehensive physical examination, which is fundamental to a physician’s diagnostic arsenal. Of course, telemedicine does not apply to every scope, such as invasive procedures, dental procedures, or critically ill patients requiring in-person visits [8]. Lack of easy access to PCPs and specialized treatment has also been associated with widespread and higher levels of perceived anxiety among patients [18]. Inadequate access to reliable information has also fostered anti-vaccine movements [19].

In an era of efforts to curb the epidemic, it is essential to safeguard the health needs of both COVID-19-infected patients and other patients. It is also important that people who identify worrisome symptoms in themselves that may indicate the development of a condition should not give up on early diagnosis [20,21]. It should also be noted that the earlier a patient is diagnosed, the greater the chances of a faster recovery, which serves to minimize treatment costs burdening the healthcare system. Therefore, it is recommended that health promotion and disease prevention activities be increased, as well as broader health education for the public for both citizens as a whole and for patients suffering from various diseases [22]. Undoubtedly, the e-health solutions implemented so far, such as e-prescription, e-referral, teleconsultation, or video consultation with a doctor, have made it possible to secure the basic needs of patients to a large extent; nevertheless, it is necessary to improve them further as doing so will make the healthcare system more resilient to emergencies (including further epidemics) in the future [23,24]. Nevertheless, when implementing such solutions, intensified information and education campaigns should also be carried out, especially those that emphasize the development of digital competencies among senior citizens burdened with multiple diseases. The elderly, for example, have repeatedly reported difficulties in using the Internet Patient Account. In the future, it should also be pointed out that, among other things, hospitals should have procedures in place to take appropriate and proportionate action, particularly about restricting the exercise of patient rights [25,26]. This restriction should not be tantamount to a ban, leading to the deprivation of patients’ rights, and should not prevent the realization of the rights of persons authorized by the patient, or relatives [27]. There is an urgent need to further standardize the provision of health services using solutions that allow remote communication [28]. Telemedicine or video consultations should not completely replace in-person highly specialized medical consultations, they should be a form of support for the patient’s treatment process in emergencies, such as in the case of the next wave of COVID-19 or the emergence of a new pandemic. However, the development of telemedicine during the pandemic was undoubtedly necessary and essential but still needs to be refined [20,22]. During the pandemic, telemedicine was an alternative method of diagnosing, treating, monitoring, and distantly supporting patients who did not require face-to-face contact with medical personnel [27,29,30]. The study conducted by the authors of this paper indicates that patients’ attitudes toward the use of telemedicine services during the COVID-19 pandemic varied. Younger people rate the quality and accessibility of teleconsultation services well, in contrast to those over 60.

### Strengths and Limitations

The study is not free of limitations. The first limitation of the conducted survey is the scope of the research sample, which includes only one specialist outpatient clinic provider from one country. However, this sample was sufficient to test and validate the research tool—a questionnaire to assess patient satisfaction with the quality of remote medical care. In addition, despite the pandemic, the survey was conducted using a face-to-face survey method, which helped reduce researcher error and the risk of “bot/fake responders”, as is the case with similar surveys conducted using the computer-assisted web interview (CAWI) method. A survey of a larger number of respondents from across the country is planned for the follow-up survey stage, which will be conducted to finalize and update the results. The second limitation is that the very evaluation of the quality of remote advice came only from the point of view of patients, who are not qualified to substantively assess the effectiveness and selection of appropriate treatment methods. The indicated research limitation provides an interesting direction for further research that could address the evaluation of the quality of the treatment by qualified medical personnel or healthcare coordinators.

## 5. Conclusions

Patients’ approach to the use of teleconsultation services during the COVID-19 pandemic varies, primarily due to attitudes toward the new situation, the age of the patient, or the need to adapt to specific solutions not always understood by the public. The availability of medical services during the COVID-19 pandemic is rated significantly lower by the elderly (over 60) and the group of pensioners/retirees. There is no gender variation in respondents’ opinions.

Telemedicine cannot completely replace inpatient services, especially among the elderly. It is necessary to refine remote visits to convince the public of this type of service. Remote visits should be refined and adapted to the needs of patients in such a way as to remove any barriers and problems arising from this type of service. This system should also be introduced as a target, providing an alternative method of inpatient services even after the pandemic ends.

## Data Availability

Not applicable.

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
