# Peer review of "Availability of Medical Services and Teleconsultation during COVID-19 Pandemic in the Opinion of Patients of Hematology Clinics—A Cross-Sectional Pilot Study (Silesia, Poland)"

_ijerph, 2023, doi:10.3390/ijerph20054264_

Round 1
Reviewer 1 Report
This study reports the results of a survey about patient opinion regarding telemedicine during the COVID-19 pandemic. It is well written and provides useful data that could help in evaluating the potential for telemedicine adoption. I think it deserves to be published if the authors provide sufficient clarifications regarding statistical analysis. Minor changes to the text are also needed.
Minor comments
Abstract: « of clinic patients » the word « clinic » could be removed
« during the pandemic well » replace with « during the pandemic as good »
« rate the availability of services during the pandemic poorly » replace with « rateD the availability of services during the pandemic AS poor » (capitals for emphasis, can be removed)
Introduction :
Could the authors provide more details about how telemedicine was implemented for most patients in the study. For example, in some cases telemedicine involves « video over IP ».
« From the standpoint of genome length and virion size, coronaviruses are some of the largest RNA viruses » : is there a reference for this assertion ?
« It is believed that people with blood type A may have an increased risk of contracting the virus and developing a severe course of the disease, while people with blood type 0 have a much lower risk » please cite the original research article (reference 2 link is dead)
Formatting issues: in my version the word in the first sentence following line 87 the word “the” appears with a strikethrough, I’m not sure if that’a local problem.
Methods:
« The probability level was set at 0.05 » : could be rephrased as « p-values <0.05 were considered statistically significant ».
More explanations are needed regarding statistical analysis. For example, according to the Methods section, the Kruskal-Wallis test and Mann Whitney U test were used throughout the article. Then, later in the article, in the sentence: “Those over 60 were more likely to negatively evaluate the availability of medical services provided during the COVID-19 pandemic (T=11.868; r= 0.632; p=0.001)“, it is unclear which test was realized. As age can be either over or under 60, the predictor is a binary variable. Therefore we expect that if questionnaire scores were considered as ordinal, a Mann Whitney test was realized, then the U statistic should be reported, not “T”. Also, the authors could explain what r= 0.632 refers to. Also, if another test was realised, the authors could explain it in the methods.
Results:
The authors should provide the absolute frequency corresponding to each percentage in the main text.
« Referring to the question: "How do you rate the availability of services provided during the COVID-19 pandemic?", a breakdown was made in the responses in terms of the number of women and men (Figure 1) » : could the authors normalise the percentages so that they add to 100 for each gender category (sum for men should be 100 and sum for women 100). Same for other questions where the answers are stratified by some category, it would be clearer if the denominator in the percentage calculation was the subcategory total rather than the grand total.
Discussion : the CAWI abbreviation should be spelled out and explained.
Conclusions : « Patients' approach to the use of teleconsultation services during the COVID-19 pandemic varies, which is primarily due to attitudes toward the new situation, the age of the patient, or the need to adapt to specific solutions not always understood by the public. » this sentence could be rephrased for clarity. I believe the author refers to « sentiment » or « opinion » rather than approach. Also, I’m not sure if « attitudes toward the new situation were evaluated in the study, instead their sentiment about specific services was evaluated.
References : some issues in references, for example reference 17 format has problems, and reference 29 could have list of authors abbreviated with « et al. »
Author Response
Dear Reviewer,
Thank you for taking the time to review your work. All suggestions have been applied, the text has been revised, the reporting of results and the language layer have been improved, the questionnaire has been added, and the manuscript has been completed with all the necessary issues identified in the review text. The changes have been marked in blue.
Reviewer 2 Report
A major comment is related with the underlying scientific hypothesis of the work done, which is not clearly started in the manuscript, somehow left to the end of the introduction.
Minor aspects to consider are:
* The introduction is too deep in coronavirus infections and should be better focused with the main focus of the article, to survey on patient's opinion on services provided during the pandemic.
*Figures 1-4 have overlap in numbers that difficult the reading.
*The reader would benefit from having access to the proprietary survey questionnaire.
Author Response

(The authors gave the same response as above.)
